

# The effectiveness of hepatitis B vaccine in toddlers based on the five-year period national basic health research (Riskesdas 2007, 2013 and 2018) in Indonesia

Christina Safira Whinie Lestari[1,*], Rita Marleta Dewi[1,*], Sunarno Sunarno[1], Armedy Ronny Hasugian[2], Sarwo Handayani[1], Masri Sembiring Maha[1], Novaria Sari Dewi Panjaitan[1], Nathalia Ningrum[3], Meiriani Sari[3] and Firda Fairuza[3]

[1] Center for Biomedical Research, Research Organization for Health, National Research and Innovation Agency (BRIN), Bogor, West Java, Indonesia
[2] Research Center for Preclinical and Clinical Medicine, Research Organization for Health, National Research and Innovation Agency (BRIN), Bogor, West Java, Indonesia
[3] Department of Paediatrics, Faculty of Medicine, Universitas Trisakti, West Jakarta, Jakarta, Indonesia
[*] These authors contributed equally to this work.

Corresponding author
Firda Fairuza, ffairuza@trisakti.ac.id

## ABSTRACT

**Background.** Hepatitis B is a viral infection that has a high prevalence in Indonesia. The Ministry of Health of Indonesia has conducted a national vaccination program for hepatitis B. In order to evaluate the success of the hepatitis B vaccination in Indonesia, a community study based on basic health research (Riskesdas) was performed nationwide since 2007 for five year period in 2007, 2013, and 2018.

**Methods.** Further statistical analysis was performed specifically for the children under 59 months old (toddlers) immunized in both urban and rural areas in 2007, 2013, and 2018 based on certain characteristics by examining antibodies against HBsAg (anti-HBs), IgG antibodies against the core antigen (HBcAb), surface antigen (HBsAg) of hepatitis B virus (HBV). The data obtained from the data management laboratory of Ministry of Health, Indonesia, was analyzed with Bivariate analysis with continuity correction chi-square or Pearson chi-square using Stata software version 16.

**Results.** This study showed an increase in hepatitis B coverage of complete immunization (30% in 2007, 60.3% in 2013, and 57% in 2018), which was also influenced by mothers' level of education (Pearson chi-square , $p < 0.05$) and access to health service points within 30 minutes (OR $= 1.3–2.8$, $p < 0.05$). The trend of the percentage of immune status (anti-HBs) was increased (41.8% in 2007; 56.1% in 2013; and 79.1% in 2018). The higher anti-HBs was found in complete hepatitis B immunization status (OR $= 1.5–2$, $p < 0.05$) and in good nutritional status ($p < 0.05$). However, the anti-HBs was found decreased with increasing age ($p < 0.05$). The trend of positive HBcAb (exposure to HBV infection) showed a decrease gradually of almost ten times from 2007 (8.6%-13.5%) compared to 2013 (2.6%-11.1%) and 2018 (1.1%-2%). Urban areas were at higher risk of hepatitis B exposure (OR $= 1.4–2.2$) than rural areas (OR $= 0.37–0.80$). The HBsAg data were only available in 2013 and 2018. Riskesdas data analysis showed the prevalence of hepatitis B (HBsAg) was lower in complete immunization status than that in incomplete one ($p < 0.05$), but with an increase from 3.9% (2013) to 9.3% (2018), possibly due to inappropriate implementation of birth dose immunization or a vaccine-escape mutant from the HBV variants.
**Conclusions.** The effectiveness of hepatitis B vaccine obtained from the three Riskesdas periods in Indonesia showed an improvement, with an increase in immune status, reduced exposure to HBV and a lower prevalence of hepatitis B in children with complete vaccination. However, there is still an increase in hepatitis B infection, especially in urban areas. Therefore, a long-term evaluation of immunization coverage especially ensuring that the initial dose of immunization was given within the first 24 h of birth, HBsAg and HBcAb, nutritional status, genomic surveillance of HBV, and other aspects of program quality evaluation are needed to ensure that elimination efforts have been implemented properly.

# INTRODUCTION

In order to evaluate the Health program in Indonesia, the Ministry of Health through the National Institute of Health Research and Development conducted basic health researches (Riskesdas) with the aim of providing health data in Indonesia and is carried out 5 years periodically, since 2007, 2013 and 2018. The prevalence of infectious diseases, the health status of infant and toddler, nutritional status, and results of serological examination of diseases preventable by immunization were included in the health data obtained through this nationwide research program. A clear description picturing of the success of the health program in Indonesia hopefully could be obtained from the implementation of Riskesdas. One of the research programs which will be evaluated and discussed in this research article is the Hepatitis B controlling program (*Kementerian Kesehatan Republik Indonesia, 2018*).

Hepatitis B is the main cause of chronic hepatitis, cirrhosis and liver cancer worldwide. World Health Organization (WHO) estimated that 296 million people were living with chronic hepatitis B infection in 2019 with 1.5 million new infections annually and an estimated 820,000 deaths, mostly from cirrhosis and hepatocellular carcinoma (primary liver cancer) (*WHO, 2022a*; *WHO, 2022b*). As an anticipation of increasing cases of chronic hepatitis and mortality due to hepatitis B, WHO in 2012 recommended implementing universal immunization coverage where 95% of children have been immunized and 180 countries have adopted this step (*Podlaha et al., 2019*; *WHO, 2012*; *WHO, 2017*). Hepatitis B is still a health problem in Indonesia, this can be seen from a study in Banjarmasin involving healthy young adults where at the time the children had not received Hepatitis B immunization, and the results obtained were that the prevalence of HBsAg, anti-HBc, and anti-HBs was nine (4.6%) respectively, 62 (31.8%), and 96 (49.2%) (*Darmawan et al., 2015*). Indonesia has carried out a national Hepatitis B immunization program since 2005 where hepatitis B immunization should be administered at 0 months and will be completed once the immunization has been given three times until the baby is 11 months old (*Kementerian Kesehatan Republik Indonesia, 2005*). Then, in order to break the chain of transmission from the mother to the infant or child, the national program for Hepatitis

B immunization administration was revised to be given at 0 day, for 4 times administration according to the announced decisions of Minister of Health (*Kementerian Kesehatan Republik Indonesia, 2015*).

The prevalence of hepatitis B in Indonesia varies from 1.9 to 11.2% according to the region and year of study (the highest is Yogyakarta 11.2% and the lowest is Bali 1.9%) (*Utsumi et al., 2014*). The prevalence rate in Indonesia is still high when being compared to several countries in Asia such as China (in children aged 1–4 years old 0.97%) (*Zhu et al., 2018*), Philippines (children aged 5–7 years 0.8%) (*Minta et al., 2021*), Lao PDR (Anti HBs 0.7%) (*Kowalczyk, Wiktor & Stepień, 2019*). Further analysis of Riskesdas data for 2007, 2013 and 2018 on serological examination of Hepatitis B was carried out to obtain an overview of the success trend of the hepatitis B immunization national program and the factors that influenced it. Furthermore, the analysis results can be used as an input for recommendations for national program in hepatitis B prevention in Indonesia.

## MATERIAL AND METHOD/METHODOLOGY

### Study design

This study is a further analysis of the data collected in Basic Health Research (Riskesdas) in 2007, 2013 and 2018, when at that time Indonesia consisted of 33 provinces (in 2007 and 2013) and 34 provinces (in 2018). Riskesdas is carried out using a cross-sectional research design, which represents community data nationally. Data was obtained from the data management laboratory of the Health Development Policy Agency, Ministry of Health, Indonesia.

### Samples

The sampling frame was designed based on calculations from the Indonesian Central Statistics Agency (BPS) by means of multistage sampling through the stages in accordance with those in the 2007, 2013, 2018 Riskesdas national reports (*Kementerian Kesehatan Republik Indonesia, 2008*; *Kementerian Kesehatan Republik Indonesia, 2013b*; *Kementerian Kesehatan Republik Indonesia, 2018*). Data from structured interviews and measurements on selected household and individual samples were then subjected to further analysis. Blood sampling and serological examination was carried out in the selected samples that were aged over 1 year old. The sub-sample used for the further analysis process was children aged 1–4 years old whom owned the complete record of immunization. The total number of samples for children aged 1–4 year old at Riskesdas 2007, 2013 and 2018 respectively are as follows 1,486 subjects, 568 subjects and 992 subjects. The 2007 sample data represents national urban areas, while 2013 and 2018 represent national urban and rural areas.

### Assessment of serological hepatitis B indicators

Blood samples for Hepatitis B serological examination were examined at the research laboratory of the Health Research and Development Agency, Ministry of Health, Indonesia. The serological examination for hepaptitis in 2007 Riskesdas was performed by using the Enzyme Linked Immunosorbant Assay (ELISA) method with Murex-Abbot laboratories reagent. Meanwhile, the 2013 and 2018 Riskesdas was performed by using the commercial

chemiluminescence immunoassay (CLIA) kit method (DiaSorin LIAISON® anti-HBs II and DiaSorin LIAISON® anti-HBc. The interpretation of the results of the serological examination is stated as follows: anti-HBs $\geq$ 10 IU/L is considered positive/protective, hepatitis B core antibody (HBcAb) at a value of <1 IU/L, hepatitis B surface antigen (HBs Ag) $\geq$ 0.05 IU/ml (CLIA), $\geq$ mean negative control + 0.05 (ELISA) is positive.

## Operational definitions

Complete hepatitis B immunization for data analysis in 2007 Riskesdas, namely hepatitis B immunization with three doses given to infants aged 0 months, 2 months and 3 months in accordance with the Regulation of the Minister of Health of the Republic of Indonesia number 53 of 2005 (*Kementerian Kesehatan Republik Indonesia, 2005*). Meanwhile, complete hepatitis B immunization for 2013 and 2018 Riskesdas, namely four doses given to infants aged 0–7 days, then continued for 2, 3, 4 months together with the DTwP vaccine in the form of the DTwP-HB combo in accordance with the 2011 Technical Guidelines which was then stipulated/written in Permenkes no 12 of 2017 (*Kementerian Kesehatan Republik Indonesia, 2017*). Meanwhile, complete hepatitis B immunization for 2013 and 2018 Riskesdas, namely four doses given to infants aged 0–7 days, then continued for 2, 3, 4 months given with the DTwP vaccine in the form of the DTwP-HB combo (2011) which was replaced by DTwP-HB-Hib pentavalent (2015) (*Kementerian Kesehatan Republik Indonesia, 2011*; *Kementerian Kesehatan Republik Indonesia, 2015*; *Kementerian Kesehatan Republik Indonesia, 2016*; *Kementerian Kesehatan Republik Indonesia, 2017*). The effectiveness of hepatitis B vaccine was based on the results of the measurements of anti-HBs, HBsAg and anti-HBc levels.

The immunization service facilities are health facilities that provide immunization services for infants and toddlers, such as hospitals, health centers, auxiliary health centers, practicing doctors, practicing midwives, integrated service posts (posyandu), village health posts, village maternity posts in villages. Economic status was measured by expenditure quintiles, namely the grouping of expenditures into five equal groups after being sorted from smallest to largest expenditure. Quintiles 1–2 are classified as poor, quintiles 3–4 as middle class and quintile 5 as rich.

The nutritional status of children under five was measured based on the weight/age index, which then converted into a standardized value ($Z$-score) using the 2005 WHO anthropometric standard (*Kementerian Kesehatan Republik Indonesia, 2011*; *WHO, 2005*). Furthermore, based on the $Z$-score value of each of these indicators, the nutritional status of children under five was determined with the following limitations: malnutrition is a combination of malnutrition and undernutrition with $Z$-score $<-2.0$, good nutrition with $Z$-score $\geq -2.0$ to $Z$-score $\leq 2.0$, and over nutrition with $Z$-score $>2.0$.

## Statistical analysis

In this further analysis, the samples taken were children aged 1–4 year old. At the time of analysis, weighting was carried out on the data every year, the value of which has been determined by the Central Agency of Statistic Republic of Indonesia. Weighting was done to ensure that there is no imbalance in the samples taken because they come from different years and so that they were surely close to the population taken.

The descriptive analysis was carried out to identify the characteristics of the respondents and factors that might influenced the effectiveness of hepatitis B vaccination using the variables available in the main Riskesdas questionnaire, namely age, sex, mother's education, hepatitis immunization status, hepatitis immunity status, respondent's travel time to the nearest health facility, and economic level. Grouping is done on several research variables to answer the purpose of further analysis. Analysis was performed using Stata software version 16. Bivariate analysis with pearson chi square was performed to assess the relationship of the research variables namely hepatitis immune status, immunization status, nutritional status and characteristics of the respondents. The significance level was 0.05 for all statistical analyses used in this study.

## Data limitation

In the 2007 Riskesdas, serology data collection was only carried out in urban areas due to limited field laboratory facilities and human resources. Not all the required variables are available in the main data, so the data analyzed is limited to the available variables.

## Ethical consideration

This studi analyzed sets of publicly available data, therefore ethical clearance was not compulsory. Permission to use and analyze the dataset was obtained by submitting a proposal to the Centre of Data and Information, Health Policy Agency (Indonesia: Pusat Data dan Informasi, Badan Kebijakan Pembangunan Kesehatan), Ministry of Health, Indonesia *via* email: datin.bkpk@kemkes.go.id.

## RESULTS

The results of the further analysis showed that in the 2013 and 2018 Riskesdas data, which included both rural and urban, most samples included were obtained from urban areas, with 50.1% and 71.6% in the respective Riskesdas periods. Meanwhile, based on gender and age, the distribution was almost even. For the education level, the highest distribution was among respondents who had graduated from high school. In both 2007 and 2018, the majority of people (49.2% and 43.8%, respectively) came from low-income families. while in 2013, the highest distribution was medium (46.7%). For travel time to immunization services, most distributions take less than 30 min. The most distribution of nutritional status is good nutrition.

For complete immunization status, there was an increase from 2007 to 2013, from 30% to 60.3%, but in 2018, there was a slight decrease to 57%. For hepatitis immune status (positive anti-HBs antibody), there was an increase from 2007 to 2018, namely 41.8%, 56%, and 79.1%. While the prevalence of hepatitis B (HBsAg) increased more than twice as much in 2013, rising from 3.9% in 2013 to 9.3% in 2018. However, the history of hepatitis B infection (HBcAb) has decreased from 10.1% in 2007 to 7.8% in 2013 and 1.3% in 2018. All the data above can be seen in Table 1. The relationship between immunization status and the education status of mothers showed that that the higher the status of mother's educational level, the higher the percentage of complete immunization status analyzed with $p < 0.05$ (Table 2).

**Table 1  Distribution of the respondents based on characteristics.**

| Characteristic | Percentage (%) | | |
|---|---|---|---|
| | Riskesdas 2007 | Riskesdas 2013 | Riskesdas 2018 |
| **Demographic** | | | |
| Area | | | |
|     Urban | 100 | 50.1 | 71.6 |
|     Rural | - | 49.9 | 28.4 |
| Sex | | | |
|     Boy (M) | 50.3 | 49.7 | 50.4 |
|     Girl (F) | 49.7 | 50.3 | 49.6 |
| Age (years old) | | | |
|     1 | 17.8 | 14.7 | 20.4 |
|     2 | 25.9 | 22.7 | 24.4 |
|     3 | 30.8 | 26.2 | 25.8 |
|     4 | 25.5 | 36.4 | 29.4 |
| Educational level of the mother | | | |
|     Unfinished Senior high school | 10.6 | 11.6 | 7.9 |
|     Senior high school | 82.8 | 83.3 | 84.2 |
|     College | 7.3 | 5.1 | 8 |
| Family economy status | | | |
|     Poor | 49.1 | 34.0 | 43.8 |
|     Middle class | 38.9 | 46.7 | 36.4 |
|     Rich | 12.0 | 19.3 | 19.8 |
| Travel time to immunization services | | | |
|     <30 min | 99.1 | 94.8 | 93.6 |
|     $\geq$ 30 min | 0.9 | 5.2 | 6.4 |
| Toddler Nutritional status | | | |
|     Undernutrition | 15.2 | 7.6 | 8.5 |
|     Good nutrition | 67.8 | 80.1 | 79.7 |
|     Overnutrition | 17.0 | 12.3 | 11.8 |
| Complete hepatitis B immunization status based on the toddler age (years old) | | | |
|     1 | 37.0 | 69.4 | 75.2 |
|     2 | 32.2 | 65.4 | 59.7 |
|     3 | 28.9 | 56.0 | 54.0 |
|     4 | 23.7 | 56.4 | 44.0 |
|     1–4 | 30.0 | 60.3 | 57.0 |
| Immune Status to Hepatitis B (Anti HBs) | | | |
|     Positive | 10.1 | 56.1 | 79.1 |
| Prevalence of hepatitis B (HBsAg) | | | |
|     Positive | - | 3.9 | 9.3 |
| Have ever been infected with Hepatitis B (Anti-HBc) | | | |
|     Positive | 0.2 | 7.8 | 1.3 |

**Table 2** Relationship between immunization status and mother's education (Riskesdas 2007, Riskesdas 2013, Riskesdas 2018).

| Survey years | Hepatitis B Immunization Status | The educational background of toddler's mother (%) | | | |
| --- | --- | --- | --- | --- | --- |
| | | Unfinished senior high school | Senior high school | College | $P^*$ value |
| Riskesdas 2007 (urban area) | Completed | 15.6 | 31.3 | 28.0 | < 0.05 |
| | Uncompleted | 84.4 | 68.7 | 72.0 | |
| Riskesdas 2013 (urban & rural areas) | Completed | 50.0 | 60.5 | 69.5 | < 0.05 |
| | Uncompleted | 50.0 | 39.5 | 30.5 | |
| Riskesdas 2018 (urban & rural areas) | Complete | 48.3 | 56.8 | 70.8 | < 0.05 |
| | Uncompleted | 51.7 | 43.2 | 29.2 | |

**Notes.**
*Pearson Chi-Square.

**Table 3** The correlation between hepatitis immunization status and respondent's travel time to immunization service facilities.

| The year of Riskesdas implementation | Area | Travel time to health facilities (minutes) | Hepatitis B Immunization Status (%) | | | | |
| --- | --- | --- | --- | --- | --- | --- | --- |
| | | | Completed | Uncompleted | $P^*$ value | OR | 95% CI |
| Riskesdas 2007 | Urban | <30 min | 29.9 | 70.1 | < 0.05 | 0.813 | 0.75–0.88 |
| | | ≥ 30 min | 34.5 | 65.5 | | | |
| | Urban and rural areas (national) | <30 min | 61.1 | 38.9 | < 0.05 | 1.883 | 1.81–1.96 |
| | | ≥ 30 min | 45.5 | 54.5 | | | |
| Riskesdas 2013 | Urban | <30 min | 61.6 | 38.4 | < 0.05 | 2.78 | 2.56–3.05 |
| | | ≥ 30 min | 36.5 | 63.5 | | | |
| | Rural | <30 min | 60.5 | 39.5 | < 0.05 | 1.68 | 1.60–1.75 |
| | | ≥ 30 min | 47.7 | 52.3 | | | |
| | Urban and rural areas (national) | <30 min | 57.5 | 42.5 | < 0.05 | 1.30 | 1.27–1.33 |
| | | ≥ 30 min | 51.0 | 49.0 | | | |
| Riskesdas 2018 | Urban | <30 min | 57.2 | 42.8 | < 0.05 | 0.61 | 0.58–0.64 |
| | | ≥ 30 min | 68.7 | 31.3 | | | |
| | Rural | <30 min | 58.4 | 41.6 | < 0.05 | 1.754 | 1.70–1.80 |
| | | ≥ 30 min | 44.5 | 55.5 | | | |

**Notes.**
*Continuity correction.

In Table 3, the relationship between hepatitis B immunization status and the length of travel time taken by respondents to immunization service facilities nationally showed that complete immunization status is higher when travel time is less than 30 min compared to travel time that is more than 30 min. All tests showed significant results ($p < 0.05$) with an increase in the completeness of immunization between 1.3 and 2.8 times (OR around 1.3−2.8) at a travel time of less than 30 min compared to a travel time of more than 30 min.

**Table 4** Effectiveness of hepatitis B immunization (anti-HBs antibody, HBsAg and HBc antibody) based on immunization status in children under five in Riskesdas 2007, 2013 and 2018 data.

| The year of Riskesdas implementation | Area | Positive markers | Hepatitis B Immunization Status (%) | | | | |
|---|---|---|---|---|---|---|---|
| | | | Completed | Uncompleted | *P* value | OR | 95% CI |
| Riskesdas 2007 | Urban | Anti-HBs | 11.4 | 8.5 | < 0.05 | 1.390 | 1.36 –1.42 |
| | | HBcAb | 0.0 | 0.1 | < 0.05 | 1.451 | 1.45 –1.46 |
| Riskesdas 2013 | Urban | Anti-HBs | 59.1 | 48.8 | < 0.05 | 1.516 | 1.47–1.56 |
| | | HBsAg | 1.0 | 2.7 | < 0.05 | 0.356 | 0.32–0.39 |
| | | HBcAb | 3.6 | 2.0 | < 0.05 | 1.806 | 1.61–2.02 |
| | Rural | Anti-HBs | 65.3 | 46.8 | < 0.05 | 2.138 | 2.08–2.20 |
| | | HBsAg | 8.9 | 3.8 | < 0.05 | 2.509 | 2.37–2.66 |
| | | HBcAb | 7.1 | 17.3 | < 0.05 | 0.366 | 0.35–0.39 |
| Riskesdas 2018 | Urban | Anti-HBs | 86.4 | 77.1 | < 0.05 | 1.893 | 1.86–1.93 |
| | | HBsAg | 7.6 | 11.1 | < 0.05 | 0.669 | 0.65–0.69 |
| | | HBcAb | 1.3 | 0.6 | < 0.05 | 2.273 | 2.08–2.29 |
| | Rural | Anti-HBs | 84.9 | 72.6 | < 0.05 | 2.127 | 2.06–2.19 |
| | | HBsAg | 9.4 | 10.9 | < 0.05 | 0.851 | 0.82–0.89 |
| | | HBcAb | 1.4 | 4.5 | < 0.05 | 0.804 | 0.76–0.85 |

Trends in the effectiveness of hepatitis B vaccination from the 2007, 2013 and 2018 Riskesdas data based on age show that the older the age, the lower the percentage of hepatitis protective antibodies (anti-HBs antibodies) (Fig. 1A). From Figure 1A, the trend of increasing hepatitis B protective antibodies was observed quite high, about 5 times and 6 times respectively from 2007 to 2013 and 2018. Meanwhile exposure to HBV, there was a trend of percentage of positive HBcAb that had decreased by almost ten times from 8.6%–13.5% in 2007, then 2.6%–11.1% in 2013, finally to 1.1%–2% in 2018 (Fig. 1B). The prevalence of hepatitis B, as assessed by positive HBsAg examination results, showed both in the 2013 and 2018 Riskesdas that the highest was in the 2-year-old group (6.3%), and there was an increase in prevalence of about 2 times, from 0.7% (6.3%) in 2013 to 5.8% (14.7%)% in 2018 (Fig. 1C).

Hepatitis B immunization immunity in the 2007, 2013 and 2018 Riskesdas data based on the completeness of hepatitis B immunization status showed that the percentage of protective immunity (anti-HBs antibody) was higher in complete immunization status than incomplete, around 1.5 to 2 times ($p < 0.05$. OR: 1.5–2), as shown in Table 4. From the 2007, 2013 and 2018 Riskesdas on positive HBc Ab data indicating that they have been exposed to hepatitis B infection, show that the risk is higher (OR: 1.4−2.2), while rural areas/ villages in 2013 and 2018 show a low risk OR: 0.366−0.804. The percentage of HBc Ab positive was lower in complete immunization status compared to incomplete in rural areas. In urban areas, however, the percentage of positive HBc-Ab is even higher with complete immunization status than with incomplete immunization status, as shown in Table 4.

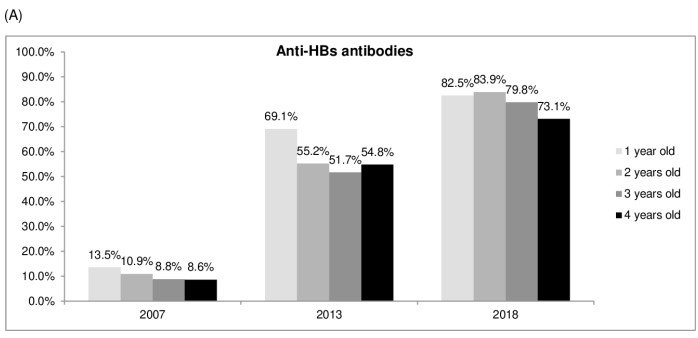

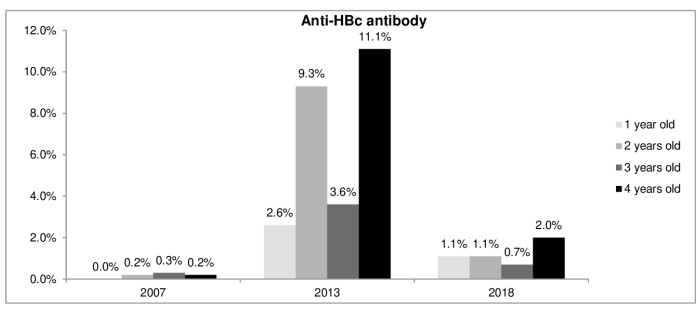

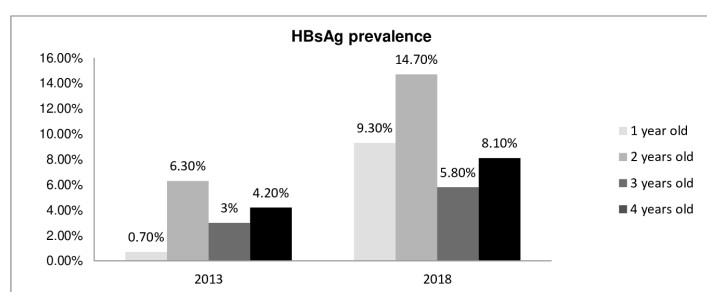

**Figure 1 Trends in the effectiveness of hepatitis B immunization (markers for anti-HBs, anti-HBc and HBsAg) based on the age of the toddler (Riskesdas 2007, 2013 and 2018) with $p < 0.05$.** (A) Percentage of protective hepatitis B immunity (anti-HBs anti body) based on age in 2007, 2013 and 2018. (B) Percentage of positive anti-HBc based on age in 2007, 2013 and 2018. (C) Percentage of positive HBsAg based on age in 2013 and 2018.

While the prevalence of hepatitis B (HBsAg positive) in urban areas is higher in the incomplete immunization status (2.7%) than in the complete (1%), On the contrary, in rural areas, the complete immunization status (8.9%) is higher than the incomplete (3.8%), while the anti-HBs is higher in the complete immunization status (65.3%) than the incomplete (46.8%). As shown in Table 4, the prevalence of hepatitis B is higher in those with an incomplete immunization status than in those with a complete one, in both urban and rural blood.

The trend of protective anti-HBs immune status for children under the age of five with complete hepatitis B immunization shows that the older the toddler, the lower the percentage of immunity, both in the 2007 and 2013 Riskesdas data and in the 2018 Riskesdas

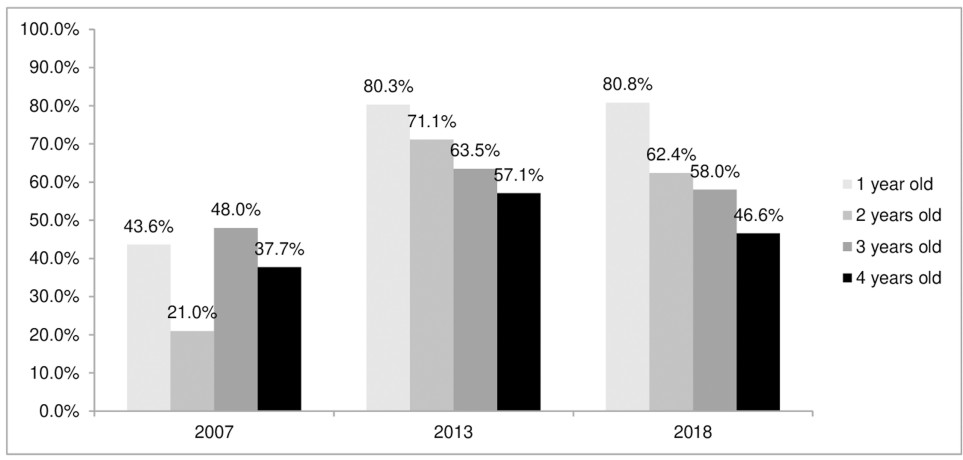

**Figure 2** Trends in hepatitis B immune status (anti-HBs) in toddlers with complete hepatitis B immunization status based on age.

**Table 5** Relationship between hepatitis B immune status (anti-HBs antibody marker) and nutritional status (Riskesdas 2007, 2013 and 2018).

| Antibody marker | The year of Riskesdas implementation | The nutritional status of toddlers (%) | | | P value |
|---|---|---|---|---|---|
| | | Undernutrition | Good nutrition | Overnutrition | |
| Anti HBs positif (%) | Riskesdas 2007 | 8.5 | 10.4 | 11 | < 0.05 |
| | Riskesdas 2013 | 44.0 | 56.1 | 56.3 | < 0.05 |
| | Riskesdas 2018 | 73.7 | 79.6 | 79.6 | < 0.05 |

data (Fig. 2). The effectiveness of the hepatitis B vaccine showed an improvement based on three periods of Riskesdas, namely with an increase in immune status (anti-HBs), a decrease in the percentage of exposure to infection (HBV) and lower HBsAg in complete hepatitis B immunization status compared to incomplete (Figs. 1A, 1B and Table 4). Based on data from the 2007, 2013, and 2018 Riskedas demonstrated that nutritional status affecting the immune system against hepatitis B, it can be seen that the percentage of positive anti-HBs in poor nutrition is lower than in good nutrition and more ($p < 0.05$), as shown in Table 5.

## DISCUSSION

The hepatitis B virus is a major cause of hepatitis infection which contributes to the high rate of morbidity and mortality. WHO has announced one dose of hepatitis B vaccination for newborns followed by two additional doses for infants to prevent vertical transmission from infected mothers to infants, thereby preventing the occurrence of chronic hepatitis B and reducing the prevalence of hepatitis B. WHO also recommends universal immunization for infants, which has a greater impact in populations with reduced hepatitis B prevalence by 34 compared to immunization of newborns born to mothers infected with hepatitis B

(*Hutin, Desai & Bulterys, 2018*). In 2016, there were only 101 countries providing routine vaccinations to all newborns; 20 countries have targeted programs, and the remaining countries do not provide newborn vaccinations (*Hutin, Desai & Bulterys, 2018*; *Li et al., 2018*). The highest proportion of babies who do not receive a birth dose is in the WHO African region (*Hutin, Desai & Bulterys, 2018*). Indonesia has implemented a universal hepatitis B vaccination program for infants since 1997, and has used a prefilled single-use injection device of hepatitis B vaccine that is stable even outside the cold chain since 1999 (*Purwono et al., 2016*). Furthermore, hepatitis B immunization has begun to be given to newborns since 2000, and has also been given a follow-up immunization schedule in the form of a combination of DTwP-HB-Hib and measles at the age of 18–24 months since 2017, according to the WHO's call to achieve the virus elimination target of hepatitis in 2030 (*Kementerian Kesehatan Republik Indonesia, 2017*).

Analyzing Riskesdas data is one way to learn about hepatitis B infection control. Some of the variables discussed in this analysis are directly related to immunization coverage and effectiveness of hepatitis B immunization, namely the education status of mothers, the distance or travel time needed to get to immunization service facilities, and the nutritional status of children. While the success of the hepatitis B immunization program is demonstrated by the frequency and effectiveness trend of hepatitis B vaccination observed by age group during the 2007–2013 and 2018–2019 Riskesdas.

Data obtained from the Riskesdas studies showed a nearly 2-fold increase in immune status (protective-level anti-HBs) in 2018 (41.8% in 2007, 56.1% in 2013, and 79.1% in 2018) (Table 1, Figs. 1A and 2), which is also in line with increasing vaccination coverage and strengthening vaccination programs. In addition, completeness of immunization status also affected the increase in the protective level of immune status (Table 4) to 1.5–2 times higher ($p < 0.05$; OR: 1.5–2), which of course is necessary to reduce HBV transmission. The immune status decreases with increasing age (Fig. 2), so booster vaccinations are still required at an older age to ensure protection against hepatitis B infection (*Bruce et al., 2016*).

The timing of the hepatitis B vaccination at the initial dose also contributed to the reduced risk of HBV exposure, which was originally given to infants from 0–1 months of age in 2005 but has been given at birth since 2015 (*Garcia et al., 2018*). In addition, the hepatitis vaccine preparation in combination with Diphteria Tetanus whole Pertusis (DTwP) or DTwP-Hib (Haemophillus Influenzae type b) vaccine provided convenience when injecting it into infants, thereby reducing the dropout rate for subsequent doses (*Kementerian Kesehatan Republik Indonesia, 2013a*; *Kementerian Kesehatan Republik Indonesia, 2015*). Although the three Riskesdas periods showed a decrease in HBcAb levels, the percentage of HBcAb was actually higher in those with complete immunization status (11.4% in 2007 compared to 3.6% in 2013 and 1.3% in 2018) compared to those with incomplete immunization status (8.5% in 2007 compared to 2% in 2013 and 0.6% in 2018). There could be other factors that affected the exposure to hepatitis B virus infection such as population density which affects the transmission of hepatitis B virus infection (*Genowska et al. , 2015*). While 2013 Riskesdas, HBsAg was found to be lower in the completed vaccinated group (1%) than that in incomplete vaccinated group (2.7%) ($p < 0.05$). However, in rural areas, HBsAg

was higher in complete immunization status (8.9%) than incomplete immunization status (3.8%), possibly due to the delay of vaccination at the initial dose for the newborn. This delay might cause the transmission from mother to infant (vertical HBV transmission) which could be the risk factor of chronic hepatitis B, for HBV infection acquired as a baby will have a risk of becoming chronic around 95%, while infection acquired as an adult has a risk of becoming chronic less than 5% (*WHO, 2022b*).

Riskesdas 2013 for children under five, which describes the immunization program 4 years earlier, starting in 2009, in rural areas, the scope of newborn vaccination may not be as wide due to geographical conditions that make it more difficult to get access to immunization services, compared to urban areas, besides strengthening the routine immunization program for all infants for hepatitis B with four new doses starting in 2015 (*Kementerian Kesehatan Republik Indonesia, 2015*). The 2018 Riskesdas data showed that the prevalence of hepatitis B (HBsAg) in both urban and rural areas was lower in the group with complete hepatitis B vaccination (7.6% urban, 9.4% rural) compared to incomplete immunization (11.1% urban, 10.9% rural, $p < 0.05$). The effectiveness of hepatitis B vaccination had shown improvement with a decreasing trend towards exposure to HBV infection (anti-HBc), lower HBV infection (HBsAg) in the fully vaccinated group and an increasing trend towards hepatitis B immune status.

Unfortunately, the prevalence of hepatitis B (HBsAg) in infants showed an increase from 3.9% in 2013 to 9.3% in 2018, which was also in line with an increase in immune status (positive anti-HBs) from 56.1% (2013) to 79.1% (2018). However, HBcAb decreased from 7.8% (2013) to 1.3% (2018) (Table 1). Positive anti-HBs indicated an immune response that is formed either due to hepatitis B vaccination or due to HBV infection. The possibility of an increase in anti-HBs indicated an increase in HBV infection, which was shown by an increase in HBsAg, because 2/3 of those infected with HBV were asymptomatic and can become carriers. This could be a source of transmission both vertically from mother to baby and horizontally from people around them who are infected with HBV (*Garcia et al., 2018*; *Genowska et al. , 2015*). This was also consistent with several other studies in Indonesia, which show that Indonesia is moderate to high endemic with an HBsAg value of around 2.5% to 10% (*Lusida & Yano, 2016*; *Wijayadi et al., 2018*; *Yano et al., 2015*). So it is very important to ensure that the first dose of immunization is given in the first 24 h when the baby is born. In addition, it could be also due to the presence of a variant HBV mutation that caused vaccine-escape mutants that had occurred in several regions in Indonesia, which had been reported in several studies. In Indonesia, four genotypes (A, B, C, D) have been identified and the most distributed are genotypes B and C. HBV subtype adw3 which is part of genotype C was a new subtype found in West Timor. Another mutation in genotype B, T126I substitution, combined with T143 found in East Java, can affect the antigenicity of HBsAg. Mutations of T140I amino acid substitution and M133T substitution, which are known as vaccine-escape mutations during immunization, also occur in unvaccinated children in Taiwan and Korea. Pre-S2 start codon mutations that occurred in Indonesia and other Asian countries were associated with chronic hepatitis (*Lusida & Yano, 2016*). The hepatitis B vaccine that has been used in the world today is the yeast-derived S-HBsAg which comes from two genotypes. The two types of vaccines have

the ability to induce strong neutralizing antibodies with a good safety profile. However, about 10% have a low or no humoral immune response in vaccinated adults. The existence of polymorphisms from the S-HBs antigenic region influenced the ability to neutralize HBV vaccine antibodies.

The government, in this case the Indonesian Ministry of Health, is also working to prevent the transmission of HBV infection from mother to child through early detection of hepatitis B from mother to child with the HBsAg rapid diagnostic which has been carried out since 2013. Starting from DKI Jakarta and continuing to other provinces the following year. Based on the Hepatitis and Gastrointestinal Infection Information System (SIHEPI) 2018–2019, the number of pregnant women tested for hepatitis B was 1,643,204 in 34 provinces. As a result, as many as 1.88% (30,965 pregnant women) were reactive (infected with hepatitis B virus), and 15,747 newborns from reactive hepatitis B mothers had been given Hepatitis B Immunoglobulin (HBIg) in less than 24 h and routine immunizations (*Kementerian Kesehatan Republik Indonesia, 2019*). Apart from that, the anti-viral drug tenofovir will also be given to pregnant women infected with HBV at 28 weeks' gestation. This program has been implemented at the Wahidin Hospital in Makassar, South Sulawesi and Karyadi Hospital in Semarang, Central Java and will be expanded to six provinces, namely DKI Jakarta, Lampung, East Java, South Kalimantan, South Sulawesi and East Nusa Tenggara (*Kementerian Kesehatan Republik Indonesia, 2016*).

The further analysis results showed that the education background of the mother has a significant effect on the completeness of the child's immunization status, including hepatitis B immunization. If the baby has received four doses of hepatitis B immunization, the immunization is complete. Complete hepatitis B immunization coverage was still low for all levels of education of mothers in 2007, but it had increased in 2013 and 2018 Riskesdas. Before 2007, DTwP and vaccine HB was initially given separately—which was given at the same time for baby aged 0, 2, and 3 months. Since mid 2015, the HB vaccine was administered four doses. The single dose purposed for newborn which was followed by the other three doses of HB vaccine combined with DTwP and *Haemophyllus influenza* type b vaccines become one-shot pentavalent vaccine (DTwP-HB-Hib) (*Kementerian Kesehatan Republik Indonesia, 2013a*). It is possible that this is related to the implementation of the new immunization program schedule, namely the first dose of hepatitis B immunization for newborns, which was announced starting in 2011.

The different number of dose of hepatitis B vaccine was given according to WHO recommendations. By distributing hepatitis B vaccine as many as three doses can provide protection from HBV infection of more than 95% in healthy infants, children and young adults (*Jain et al., 2021*). Vaccination can also reduce HBV infection in newborns of infected mothers by 3.5 times. However, the efficacy of the vaccine in preventing perinatal transmission decreases as the length of time the first dose is administered at birth increases (*Potsch et al., 2010*). The risk of HBV infection in newborns was found to be 8 times higher when the first dose was given after 7 days of birth compared with giving within 3 days after birth (*Jain et al., 2021*; *Potsch et al., 2010*). In areas where chronic infection is endemic, timing of the first dose of hepatitis B vaccine is critical with minimal delay after birth. In countries with low endemicity, administration of the first dose at birth

is also necessary to prevent early transmission. Resolution 63.18 of the World Health Assembly, which focused on viral hepatitis, recommends that all infants in the world receive the first dose of hepatitis B vaccine immediately after birth, preferably within 24 h (*World Health Organization, 2013*). Several other studies have also shown that the average seroprotective rate is higher in those given four doses of the HB vaccine compared to three doses in both children and adults (*Jain et al., 2021*; *Potsch et al., 2010*). Because Indonesia is included in an area with a high prevalence, the program is carried out according to WHO recommendations, by carrying out four doses of vaccination to prevent transmission of HBV infection as early as possible and ensure a person's protection against horizontal transmission (*World Health Organization, 2013*).

Concerning mother education and the completeness of babies receiving complete basic immunizations, including hepatitis B, several studies in Indonesia show that knowledge and education of mothers have a significant impact on child immunization compliance and completeness (*Kibreab, Lewycka & Tewelde, 2020*; *Nanda Kharin et al., 2021*). Studies in several countries in Africa, such as Nigeria and Eritrea, also show that mothers who are literate and educated will have better reading skills and health-seeking behaviors, including seeking immunization (*Balogun et al., 2017*; *Kibreab, Lewycka & Tewelde, 2020*). Apart from being influenced by the mother's education, immunization status is influenced by the household wealth index, the mother's ANC visits, region, and ownership of the vaccination card (*Kibreab, Lewycka & Tewelde, 2020*). A literature study conducted by *Kaufman et al. (2018)* shows that face-to-face education about immunization can improve immunization status in infants. This is related to the increased knowledge, attitudes, and motivation of mothers to immunize their babies. The same thing happens if the mother receives immunization education during her pregnancy (*Saitoh et al., 2013*). Ignorance about the need for immunization is the main cause of an incomplete or even unimmunized children's immunization status (*Ratta & Meshram, 2020*).

In addition to the educational status of the mother, the access to health care facilities as defined by travel time, both in urban and rural areas also showed a significant effect on the completeness of hepatitis B immunization status, based on the data of 2007, 2013 and 2018 Riskesdas. This travel time is related to the distance to where you live and the ease of transportation to health services. The 2013 Riskesdas analysis shows that proximity, convenience, and knowledge about access to health services affect the completeness of a child's immunization status (*Nainggolan, Hapsari & Indrawati, 2016*). One of the efforts of Indonesian government to facilitate access to community-based health services is to establish posyandu, poskesdes, and polindes, one of whose health services is immunization. This is very useful, especially for rural areas. In countries in the poor-middle category, the factor of ease of access to health services plays a very important role, the same is also found in rural India (*Ratta & Meshram, 2020*). However, based on an analysis of immunization coverage data from 2013 on 45,095 children aged 2–3 years in China, the maternal education and access to health services was shown to have no effect on children's complete immunization status. Other factors that actually have an effect are: sons, a father's education that did not finish high school, being born in a house, and living in a suburb or mountain area (*Cao et al., 2018*). The existence of differences in the

dominant factors affecting the completeness of immunization status in several countries may be due to, among other things, the availability of health facilitation, socio-economic differences, and cultural and geographical cultures. In Sub-Saharan Africa, based on 23 demographic and health surveys involving 26,241 children between 2010 and 2018, it shows that the existence of a socio-economic gap between urban and rural areas greatly influences children's immunization status. Another factor is the number of births and the distance to health facilities (*Ameyaw et al., 2021*).

The nutritional status was one of important factors affecting the immune system against hepatitis B in this study based on the data obtained from 2007, 2013, and 2018 Riskesdas. Besides, nutritional deficiencies were reported to strongly affect patients diagnosed with liver diseases, such as HBV-caused hepatitis and cirrhosis, especially in children. *Yu et al. (2017)* had examined and reported prevalence of malnutrition and malnutrition risks in the hospitalized children with liver diseases. Previous study which analyzed the 2013 Riskesdas data had highlighted and well discussed that the failure in to get sufficient nutrition intake was related to challenging life environment, socioeconomic status, and infectious diseases in children under five year old (*Kusrini, Mulyantoro & Supadmi, 2020*). Meanwhile, another previous study comparing the progression of cirrhosis caused by hepatitis B virus infection in malnourished patients and nourished patients reported that there were no significant difference in liver-specific complications, such as variceal bleeding, ascites, and hepatic encephalophathy in these two different nutrional status groups of patients (*Chen et al., 2021*). However, the relationship between hepatitis vaccine effectiveness and nutritional status in children was still unraveled. Based on our knowledge and readings, there was only one previous study reported the antibody responses toward hepatitis B vaccine in adult patients under hemodialysis (*Afsar, 2013*). The nutritional and immunological factors were also assayed in this previous study and shown to positively influence the seroconversion towards hepatitis B vaccination.

Defects in cellular and humoral immunity as well as phagocyte function disorders can be caused by protein energy malnutrition (PEM). In addition, reductions in the levels of the complement (except C4), secretory immunoglobulin A and cytokines can result. Malnutrition, secondary to deficiencies in proteins, metal elements or vitamins, can lead to changes and severe atrophy in the thymus gland because of apoptosis-induced thymocyte depletion. This can lead to a persistent lymphocyte reduction, particularly in the immature CD4þ CD8þ cells, and to a decrease in cell proliferation. It is noteworthy that these conditions can be reversed following proper diet rehabilitation (*Savino, 2002*). PEM also reduces the concentrations of IgA, IgM and IgG and the production of cytokines (*Scrimshaw, & SanGiovanni, 1997*). PEM-associated micronutrient deficiencies also have adverse effects on the immune response. Deficiency in zinc, iron, copper, vitamins plays an important role in reduced immune response as a result of malnutrition (*Keusch, 2003*). Zinc, copper and selenium are needed to maintain and reinforce immune and antioxidant systems (*Mocchegiani et al., 2014*). Vitamin A stimulates cell differentiation and cytokine secretion by macrophages, including tumor necrosis factor, interleukin (IL)-1, IL-6, and IL-12 (*Mohty et al., 2003*).

In this study based on the data obtained from 2007, 2013, and 2018 Riskesdas demonstrated that nutritional status was one of important factors affecting the immune system against hepatitis B. It can be seen that the percentage of positive anti-HBs in malnourish children is lower than in normal nutritional status ($p < 0.05$), as shown in Table 5. This is similar with a study in Senegal and Cameroon, they found the percentages of protected children were lower among children with moderate to severe malnutrition *versus* normal nutritional status (60% *vs* 85%) and nutritional status was significantly correlated with the response to HBV vaccination ($p < 0.001$) (*Rey-Cuille et al., 2012*). In Iran, *Karimi et al. (2013)* found a significant decrease in the immune response to HBV vaccination in malnutrition children (60.2%). Nutritional status is major influencing factor in immunologic response; and is major factor of immunodeficiency. Protein energy malnutrition (PEM) was found to cause a decrease in cellular and humoral immunity and phagocyte function disorders; complement level (except C4), secretary IgA, and cytokine production (*Cantani, 2000*; *Keusch, 2003*; *VanLoveren et al., 2001*). PEM also reduces the concentrations of IgA, IgM and IgG and the production of cytokines (*Scrimshaw, & SanGiovanni, 1997*). Micronutrient deficiency in zinc, selenium, Fe, copper, vitamins A, B, C, E, B6, and Folic acid play important role on the immune response in malnutrition (*Mocchegiani et al., 2014*; *Scrimshaw, & SanGiovanni, 1997*). This study showed a significant association between anti-HBs levels and the normal nutritional status, the higher anti-HBs was found in normal nutritional status ($p < 0.05$). Protective rates in children with malnutrition *versus* normal nutrition based on the data obtained from 2007, 2013, and 2018 was 8.5%, 44%, 73.7% *vs* 10,4%, 56,1%, 79,6%, respectively, with a statistically significant difference ($p < 0.05$) (Table 5).

The 2007 Riskesdas study, conducted in 21 of 33 provinces in Indonesia, found an HBsAg prevalence of 9.4% in all age groups (*Muljono, 2017*), which dropped nearly threefold to 3.9% in 2013 and then rose again to 9.3% in 2018 (Table 1, Fig. 1C). The results of the 2018 Riskesdas are almost identical to the meta-analysis of 26 cohort studies (20 studies from China, 2 each from Gambia and Italy, and 1 each from Australia and Fiji) which assessed the long-term impact of hepatitis immunization. When the prevalence of hepatitis B infection is compared to the prevalence of HBV infection (HBsAg positive or HBc positive or both are positive), the unvaccinated group showed a prevalence of 0.6%–16%, while the vaccinated group showed a prevalence of 0.3−8.5% (*Whitford et al., 2018*). However, Table 1 and Fig. 1B showed a significant decrease of almost 10 times for HBc Ab positives (10.1% in 2007; 7.8% in 2013; and 1.3% in 2018). This result indicated that the immunization program in Indonesia has succeeded in reducing transmission of hepatitis B both vertically from mother to baby, as well as horizontally or in the surroundings because immunity has been formed, which is quite good according to the results of studies on immune status in the 3 periods of Riskesdas. This result is in accordance with the improvement in the policy for the hepatitis B vaccination program for all infants in Indonesia which has an increasingly wider coverage, which was originally given in 2005 with 3 doses, then since 2015 it has become four doses (*Kementerian Kesehatan Republik Indonesia, 2005*; *Kementerian Kesehatan Republik Indonesia, 2015*).

## CONCLUSIONS

There was a fairly good increase in complete immunization status (from 30% to 57%) and immune/anti-HBs status (from 41.8% to 79.1%) with exposure to HBV (HBc Ab), which decreased significantly from 10.1% to 1.3% over the three Riskesdas periods in 2007, 2013, and 2018. This indicateed the effectiveness of the hepatitis B vaccine was also getting better. The higher immunization coverage was also influenced by the higher education level of the mother and the travel time to the immunization service which was less than 30 min. Meanwhile, the better immune response is influenced by good nutritional status. The routine hepatitis B immunization program for all babies in Indonesia has been getting better with the provision of vaccination to newborns, four doses with a combination preparation with the DTwP vaccine and Hib vaccine, which has been implemented since 2015, and an increasingly wide program coverage area in Indonesia. However, there were still hepatitis B infection cases reported that vary from low to high. Therefore, long-term evaluation of HBV prevalence (HBsAg and HBcAb), immunization coverage, immune status, nutritional status, timeliness of vaccine delivery schedules especially the accuracy of giving the first dose before 24 h after birth, and other aspects of program quality are surely needed. To ensure that elimination efforts have been carried out correctly, adolescents and young adults with a history of vaccination must be evaluated.

## ACKNOWLEDGEMENTS

We would like to acknowledge the Head of the Health Development Policy Agency, the Indonesian Ministry of Health and the data management laboratory team, who had given permission to use the data and provided the 2007, 2013 and 2018 Riskesdas research data sets. In addition, gratitude was conveyed to all the research teams who cannot be mentioned one by one, specifically from the Health Research and Development Agency, Ministry of Health of Indonesia, whom had collected data on Riskesdas research, laboratory examinations, and data management processes.

### Funding
This work was supported by the grant provided by the Faculty of Medicine, Universitas Trisakti, Jakarta, Indonesia (No. 0388/PUF/FK-2022-2023). The funders had no role in study design, data collection and analysis, decision to publish, or preparation of the manuscript.

### Grant Disclosures
The following grant information was disclosed by the authors:
Faculty of Medicine, Universitas Trisakti, Jakarta, Indonesia: 0388/PUF/FK-2022-2023.

### Competing Interests
The authors declare there are no competing interests.

## Author Contributions

- Christina Safira Whinie Lestari, Rita Marleta Dewi, Sunarno Sunarno and Armedy Ronny Hasugian conceived and designed the experiments, performed the experiments, analyzed the data, prepared figures and/or tables, authored or reviewed drafts of the article, and approved the final draft.
- Sarwo Handayani, Masri Sembiring Maha and Novaria Sari Dewi Panjaitan analyzed the data, prepared figures and/or tables, authored or reviewed drafts of the article, and approved the final draft.
- Nathalia Ningrum, Meiriani Sari and Firda Fairuza analyzed the data, authored or reviewed drafts of the article, and approved the final draft.

## Human Ethics

The following information was supplied relating to ethical approvals (i.e., approving body and any reference numbers):

This research received ethical approval from the Research Ethics Commission of the Faculty of Medicine, Trisakti University, numbered: 164/KER/FK/VIII/2022 dated 22 August 2022.

## Data Availability

The data are available upon request to the Center of Data and Information, Health Policy Agency (Indonesia: Pusat Data dan Informasi, Badan Kebijakan Pembangunan Kesehatan), Ministry of Health, Indonesia at https://www.badankebijakan.kemkes.go.id/.

Select "Menu Layanan Data" (Data service menu), select "Tata Cara Permintaan Data" (Data request procedures). Instructions are at: "Prosedur Pengajuan Permintaan Data Hasil Litbangkes" (Procedures and requirements for Submitting Requests for Data on Research and Development Results).

The request must be submitted with a cover letter signed by the head of the institution, Data Request Submission Form, and project proposal and emailed to datin.bkpk@kemkes.go.id.

## Supplemental Information

Supplemental information for this article can be found online at http://dx.doi.org/10.7717/peerj.15199#supplemental-information.

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
