# Peer review of "The effectiveness of hepatitis B vaccine in toddlers based on the five-year period national basic health research (Riskesdas 2007, 2013 and 2018) in Indonesia"

_PeerJ, doi:10.7717/peerj.15199_

## Round 0.1 · original submission · Major Revisions

Dear Dr. Fairuza,

Thank you for your submission to PeerJ. I am pleased to inform you that
the manuscript titled "The effectiveness of hepatitis B vaccine in toddlers based on the five-year period national basic health research (Riskesdas 2007, 2013 and 2018) in Indonesia' has been reviewed by the reviewers. On the basis of reviewers' reports, the manuscript has been recommended for major revision. Please address the major and minor issues and resubmit the revised manuscript with point wise answers.

Reviewer 1 ·

Basic reporting

1. What was the DPT vaccine? All abbreviations should be spelt out whenever first mentioned.
2. The language of this paper should be carefully checked and edited.
3. In the Discussion section, “Only 101 countries provide routine vaccinations to all newborns; 20 countries have targeted programs, and the remaining countries do not provide newborn vaccinations.” Was there any reference or evidence to support this statement? If true, in which year?

Experimental design

1. In the Samples section, “In the selected sub-sample, children aged over 1 year were taken for blood sampling and serological examination was carried out for their serum. The sample used for the further analysis process was children aged 1-4 years who were a sub-sample and included in the group that underwent serological examination.” This was confusing. At first it seems children under 1 year were included in the selected samples but then children aged 1-4 years were selected. How were the samples selected and what were the sub-samples and what were they used for?
2. In the Statistical Analysis section, the authors stated that “In this further analysis, the samples taken were toddlers (children aged <= 59 months).” However, previous section stated that children aged 1-4 years were selected for analysis. Exactly how were the samples/data selected for analysis? Please clarify.

Validity of the findings

1. Different numbers of doses of vaccines were given at different times between 2007 and 2013,2018 (3 doses at 0,2,3 months vs. 4 doses at 0-7 days, 2,3,4 months). This could be a potential cause of difference and should be studied and discussed.
2. P-values being exactly 0 is not possible.
3. In the Discussion section, “Some of the variables discussed in this analysis are directly related to immunization coverage and effectiveness of hepatitis B immunization, namely the education status of mothers, the distance or travel time needed to get to immunization service facilities, and the nutritional status of toddlers.” Household income was included in the analysis which could be a confounder for “immunization coverage” and “the education status of mothers, the distance or travel time needed to get to immunization service facilities, and the nutritional status of toddlers”. Did the authors have any adjustment for confounding?

Reviewer 2 ·

Basic reporting

It is descriptive study which analyze the serological data for prevalence of hepatitis B in Indonesia. The study shows increase in HBV vaccine coverage over the years starting from 2007 to 2018. All parameters e.g. antibodies status to various HBV antigens, mothers' level of education etc. indicate to validate the findings. However, despite better coverage an increase in disease prevalence from 2013 to 2018 was reported for which there is no defined explanation.

Experimental design

The methods used are appropriate and sufficient.

Validity of the findings

Are appropriate and sufficient.

Additional comments

I would suggest if authors can further comment on
i) Why there was an increase in disease prevalence from 2013 to 2018 despite better immunization coverage. Can the data be authenticated by some studies.
ii) From the MS I have not been able to understand much about the impact improved nutritional status has made on enhanced immune response.

---

## Round 0.2 · accepted · Accept

The authors have addressed all the reviewers' comments in the revised manuscript. I assessed the revision and found that the current version of the manuscript is acceptable for publication.

Reviewer 1 ·

Basic reporting

No comment.

Experimental design

No comment.

Validity of the findings

No comment.